# Predicting the Level of Respiratory Support in COVID-19 Patients Using Machine Learning

**DOI:** 10.3390/bioengineering9100536

**Published:** 2022-10-09

**Authors:** Hisham Abdeltawab, Fahmi Khalifa, Yaser ElNakieb, Ahmed Elnakib, Fatma Taher, Norah Saleh Alghamdi, Harpal Singh Sandhu, Ayman El-Baz

**Affiliations:** 1Department of Bioengineering, University of Louisville, Louisville, KY 40292, USA; 2College of Technological Innovation, Zayed University, Dubai P.O. Box 19282, United Arab Emirates; 3Department of Computer Sciences, College of Computer and Information Sciences, Princess Nourah Bint Abdulrahman University, P.O. Box 84428, Riyadh 11671, Saudi Arabia

**Keywords:** COVID-19, respiratory support, machine learning, feature selection

## Abstract

In this paper, a machine learning-based system for the prediction of the required level of respiratory support in COVID-19 patients is proposed. The level of respiratory support is divided into three classes: class 0 which refers to minimal support, class 1 which refers to non-invasive support, and class 2 which refers to invasive support. A two-stage classification system is built. First, the classification between class 0 and others is performed. Then, the classification between class 1 and class 2 is performed. The system is built using a dataset collected retrospectively from 3491 patients admitted to tertiary care hospitals at the University of Louisville Medical Center. The use of the feature selection method based on analysis of variance is demonstrated in the paper. Furthermore, a dimensionality reduction method called principal component analysis is used. XGBoost classifier achieves the best classification accuracy (84%) in the first stage. It also achieved optimal performance in the second stage, with a classification accuracy of 83%.

## 1. Introduction

In December 2019, a group of atypical pneumonia presented in Wuhan, China [1]. Subsequently, the National Health Commission (NHC) of the People’s Republic of China declared that a novel coronavirus is responsible for the outbreak [2]. The novel virus was named coronavirus disease 2019 (COVID-19) by the World Health Organization (WHO). High throughput sequencing resulted in considering COVID-19 as a betacoronavirus. COVID-19 is genetically similar to the coronaviruses found in bats, furthermore, it shares about 50% and 79% of its genetic sequence with the coronaviruses responsible for the Middle East respiratory syndrome (MERS) and severe acute respiratory syndrome (SARS), respectively [3]. Despite the epidemiological evidence which suggests that the majority of the initial patients visited the Huanan Seafood Market in Wuhan, the zoonotic origin of COVID-19 still was not identified. Now, most new infections come from human-to-human transmissions, including those among health care workers and family members [4,5,6]. As of 2 February 2022, there are about 380 M confirmed cases and there are about 5.7 M confirmed deaths [7] worldwide.

The most frequent serious clinical manifestation of the disease is viral pneumonia. The features of the pneumonia are fever, cough, dyspnea, hypoxemia, and bilateral infiltrates on chest radiography [1,5,6,8]. It is more common that the patient has dry cough than a productive cough [5]. After a median time of five to eight days, dyspnea appears [1,5]. Severe hypoxemic respiratory failure appears in a significant proportion of the patients with COVID-19 pneumonia [9,10]. A high risk of death is associated with patients who require mechanical ventilation [11]. We can conclude that COVID-19 is associated with respiratory disorders and respiratory support is required for patients with severe disease.

Significant long-term symptoms from COVID-19 (post-acute sequelae COVID-19 syndrome) affect approximately 30–50% of patients hospitalized with acute COVID-19 infection. Predicting which patients will need respiratory support is an active area of research with important implications for ensuring appropriate patient care, follow-up, and healthcare resource allocation [12]. The required patient respiratory support can be classified into three levels minimal, non-invasive (high-flow nasal cannula or CPAP), and invasive mechanical ventilation.

Machine learning methods are increasingly utilized to enhance the prediction of respiratory failure. Badnjevic et al. [13] implemented a method for the classification of asthma and chronic obstructive pulmonary disease. Their method was based on fuzzy rules and the trained neural network. In the classification between the two classes, a sensitivity of 99.28% and a specificity of 100% were achieved. In another work by Badnjevic et al. [14], the same two types of respiratory diseases were classified using an expert diagnostic system that was validated on a higher number of patients. The system reached a sensitivity of 96.45% and a specificity of 98.71%. Nopour et al. [15] used machine learning algorithms to predict the need of intubation in a small number of COVID-19 patients (482). The best performing machine learning model yielded an area under the curve of 0.892. Kabbaha et al. [16] investigated the association between several factors and the need for invasive mechanical ventilation using a large population from the middle east. They used machine learning approaches and achieved an area under the curve of 0.718. Nirmaladevi et al. [17] used an unsupervised deep convolutional neural network to classify COVID-19 patients into four classes according to the seriousness of the disease using chest X-ray images. An accuracy of 96% was achieved. Zeidberg et al. [18] proposed two machine learning methods: logistic regression and XGBoost. They applied the methods to patients that suffer from acute respiratory distress syndrome. The best area under the ROC curve (0.81) was obtained from L2 logistic regression. Many methods have been proposed specifically for COVID-19 patients. Ferrari et al. [19] proposed a method that combines machine learning algorithms such as ensemble decision trees with the experience of doctors to predict forty-eight hours in advance which patient will develop moderately-severe respiratory failure. They achieved a predictive accuracy of 84%. Burdick et al. [20] utilized XGBoost for fitting decision trees to predict patients who will need invasive mechanical ventilation and they reached an area under the curve of 0.87. Bolourani et al. [21] proposed a model that is based on XGBoost and reached an area under the curve of 0.77. Recently, Bendavid et al. [22] proposed an approach to predict the need for invasive mechanical ventilation in COVID-19 patients reaching an area under the curve of 0.97. Our work is the first study that proposes an automated method to predict the level of respiratory support in terms of three classes. Therefore, our method introduces a more accurate grading of the level of respiratory support.

In this paper, a machine learning based framework that predicts the clinical severity of the disease was proposed. The severity of the disease is defined in terms of the amount of respiratory support the patient will require (e.g., ventilation support) using clinical numerical data collected from patients admitted to tertiary care hospitals. The proposed system is intended to help streamline physicians’ decision making with regard to immediate care of COVID-19 patients. Unlike the literature work, e.g., [20,22], that attempted to detect whether the COVID-19 patient would require invasive mechanical ventilation only, here a classification problem with three classes (0 minimal support, 1 non-invasive, and 2 invasive) was investigated. To the best of our knowledge, this work is the first study to investigate this problem using machine learning approaches.

## 2. Materials and Methods

A method for predicting the required level of respiratory support in COVID-19 patients using machine learning algorithms was proposed. The proposed framework of our work is shown in Figure 1. The following sections explain each part of the framework. Initially, one stage classification using similar algorithms as presented in the manuscript was investigated. However, a low classification accuracy of 73% was obtained. Therefore, a two stages system was used for classification which improved the accuracy. The three investigated classes are not classified in one step, instead, the system has two stages. Class 1 with class 2 were combined as one class. Therefore, in the first stage, our system is able to differentiate between class 0 and the others. Then, in the second stage, our system is able to classify between class 1 and class 2. The ultimate goal of the study is to help clinicians in decision making regarding ventilation requirements of COVID-19 patients. All of our work has been done using Python programming language and its libraries Numpy, Pandas, and Scikit-Learn [23].

### 2.1. Data Description

Retrospective data were collected by the Center of Excellence in Respiratory Infectious Diseases at the University of Louisville (UofL) starting in February 2020, and the data analyzed here extend from the creation of the dataset through 31 December 2020. Eight different hospitals in the Louisville metropolitan area were included, constituting one academic medical center, two other tertiary care hospitals, and five community hospitals in the area. Inclusion criteria were a positive COVID-19 polymerase chain reaction test, symptomatic infection, and admission to the hospital. Symptoms were defined as any respiratory symptoms, diarrhea, or those of systemic sepsis. The only exclusion criterion was age less than 18. Complied with the Helsinki Declaration, the acquisition and data collection were approved by the UofL Institutional Review Board (IRB); the IRB number is 21.0673. The data set contains 3491 patients with 16 raw features, namely, height, weight, systolic blood pressure, diastolic blood pressure, heart rate, body temperature, respiratory rate, oxygen saturation, fraction of inspired oxygen, pulmonary arterial pressure, platelet count, lymphocyte count, neutrophil count, C-reactive protein, lactate dehydrogenase, and D-dimer. The features were taken from a single timepoint, which is the initial presentation of patients to the hospital. This is either the measurement taken in the emergency room or the first measurement taken upon admission to the inpatient ward.

### 2.2. Data Preparation

In our work, the data were prepared using the following modules:

#### 2.2.1. Data Wrangling

Data wrangling, also known as data cleaning, stands for a set of processes designed to convert raw data into more easily utilized formats. The exact processes differ according to the project and the data which can be leveraged. In our work, three processes of data wrangling was used. First, the features were renamed into more ready-to-use formats. Renaming is important to facilitate data analysis using the pandas tool. Second, outliers were removed from each feature. An outlier is a data point that differs significantly from other observations. It may be due to variability in the measurement or it may refer to experimental error; the latter are sometimes excluded from the data set. Outliers can harm the performance of the machine learning model and produce statistically insignificant results. Therefore, removing them is of great importance. Finally, data imputations were performed for all features that contained nan values. In our work, imputation by the mean value was used.

In the outlier removal step, the entire record is eliminated from the feature matrix. The initial number of patients/records was 3491 and after outlier removal, 2962 observations remained. Thus, a high number of outliers (529) was found in the dataset. Examples of outliers were RR = 0 breathes/minute and HR = 1302 beats/minute. These values are clearly spurious and only present due to value insertion errors. The machine learning system was built using the remained 2962 observations. On the other hand, the dataset contained nan values. Therefore, imputation by means was used to fill these gaps. To check the validity of the feature values, summary statistics were generated for each feature and the summary values were reasonable and logical.

After data preparation step, our feature matrix had a size of (2962 observations, 12 features). Then, a two-stage classification system was built. In the first stage, the system differentiates between class 0 and other classes (class 1 and class 2). In the second stage, the system differentiates between class 1 and class 2. Thus, a binary classification was used in each stage.

#### 2.2.2. Feature Engineering

Feature engineering refers to the methodology of applying domain knowledge to extract new and useful variables from raw data during the creation of a predictive model using the algorithms of machine learning. The ultimate goal of feature engineering is to enhance the performance of machine learning algorithms. In our work, four new features were extracted from the raw features, where the final number of features equals twelve features. The body mass index (BMI) feature was created by using Equation (Equation 1). The mean arterial pressure (MAP) feature was created using Equation (Equation 2). Ratio of pulmonary arterial oxygen to fraction of inspired oxygen was used as a feature and we named it PaO2/FIO2. It is worth mentioning that PaO2 is not directly recorded, however, it has been estimated from the percent oxygen saturation in the blood using various tables and graphs [24]. Finally, a feature that is described as the ratio between neutrophil count and lymphocyte count was created. The engineered features were created after recommendations from the clinical collaborator.
(1)$$BMI=\frac{Body\;Mass\;(kg)}{{Height}^{2}}$$
(2)$$MAP=\left(\frac{2}{3}\right)*Diastolic\;Pressure+\left(\frac{1}{3}\right)*Systolic\;Pressure$$

The features were renamed with small names that do not contain spaces. In addition, four features were engineered from the raw features. Thus, the total number of features is 12 with the following names: BMI, MAP, HR, T, RR, PaO2/FiO2, platelets, lymphocytes, neutrophils, CRP, LDH, and D-dimer which stand for body mass index, mean arterial pressure, heart rate, body temperature, respiratory rate, ratio of pulmonary arterial oxygen to fraction of inspired oxygen, platelet count, lymphocytes count, ratio of neutrophils to lymphocytes, C reactive protein, lactate dehydrogenase, and D-dimer. The mean and standard deviation of each feature across the three classes are presented in Table 1. The classes have different features means.

#### 2.2.3. Feature Selection and Dimensionality Reduction

Feature selection is the process of decreasing the number of features by selecting the most relevant and non-redundant ones when developing a predictive model. Reducing the number of input features can decrease the computational cost as well as improve the performance of the model, in some cases. The types of feature selection methods can be summarized to unsupervised and supervised. In this work, the performances of two feature selection methods were evaluated, separately. A supervised method called filter-based method with statistical measure (analysis of variance (ANOVA)) as well as the recursive feature elimination method were used. Besides feature selection methods, a dimensionality reduction method, called principal component analysis (PCA) [25] was investigated. ANOVA and PCA provided us with the best classification accuracies. This is because the recursive feature elimination might exacerbate overfitting. Therefore, in this paper, the results from ANOVA and PCA were presented.

In filter-based selection method, the relationship between each input feature and the target is evaluated using statistical techniques. Then, our basis is the estimated scores to filter those input features which will be considered in the modeling. Correlation type statistical measures are commonly used as the basis when evaluating the relationship between the input and output variables. Because the paper solves a classification problem with numerical input and categorical output, ANOVA correlation coefficient [26] was used in our filter-based feature selection method. The assumptions of ANOVA are satisfied in our work. Please refer to Algorithm 1 for an overall idea about the basic steps of selecting the best features for optimal model performance. Our selection criterion is based on the accuracy of the validation set.

Additionally, PCA was used to reduce the dimensions of the feature matrix before feeding it to the machine learning model. In PCA, the principal components are computed and they are used to make a change in the basis of the data. Usually, the first few components are used and the rest is ignored. PCA projects each data item onto the first few components to get a lower-dimensional data while retaining as much of the variation of the data as possible. Please refer to Algorithm 2 for an overall idea about the basic steps of selecting the best configuration using PCA for optimal model performance. Our reduction criterion is based on the accuracy of the validation set.
**Algorithm 1** Basic steps for determining the best training settings for the model using ANOVA.accuracies = []                               ▹ Empty list**for**n = 1 to N**do**                       ▹ N is the number of features      Determine the best n features using ANOVA.      Discard other features.      Train the model using the training set.      Evaluate the model using the validation set.      accuracies[n] = accuracy of the model on the validation set.**end for**choose the best configuration based on the accuracy maximum value.Evaluate the model using the testing set.

**Algorithm 2** Basic steps for determining the best training settings for the model using PCA.
accuracies = []                               ▹ Empty list
**for**n = 1 to N**do**                       ▹ N is the number of features
      Resize feature matrix to (observations,n) by PCA
      Train the model using the training set.
      Evaluate the model using the validation set.
      accuracies[n] = accuracy of the model on the validation set.

**end for**


choose the best configuration based on the accuracy maximum value.


Evaluate the model using the testing set.



#### 2.2.4. Feature Scaling

Standardization of the features was performed by removing the mean and scaling to unit variance. For a sample *x*, the standard score is calculated as:(3)$$z=\frac{\left(x-u\right)}{s}$$
where *u* stands for the mean of the samples, and *s* stands for the standard deviation of the samples.

### 2.3. Machine Learning Models

Five machine learning models were used in this paper, namely, logistic regression (LR), random forest (RF), support vector machine (SVM), multi-layer perceptron (MLP), and XGBoost classifiers. For 10 different iterations, the dataset was randomly divided into a training set, validation set, and testing set with proportions that are equal to 50%, 25%, and 25%, respectively. Hyper-parameters tuning of the models is very important to obtain optimal performance. Therefore, a grid search was used with each model to find the best parameters for it. The accuracy of the validation set was our criterion that determines the selection of best parameters as shown in Algorithms 1 and 2.

#### 2.3.1. Logistic Regression

One of the most popular algorithms that are utilized to solve binary classification tasks is the logistic regression (LR). It is considered a supervised learning approach that can be used when the labels are either 0 or 1, as shown in Equation (Equation 4):(4)$$\hat{y}=P\left(y=1|x\right)x\in R^{n_{x}}$$
where $\hat{y}$ is the chance of y=1, given the input features *x*, and the *x* is an $n_{x}$—dimensional vector. *w* refer to the parameters of logistic regression, which is also an $n_{x}$—dimensional vector together with *b* as a real number. Now, given an input *x* and the parameters *w* and *b*, the output can be generated using linear function. However, this is not a good algorithm, because we want $\hat{y}$ to be the chance that y=1. Therefore, $\hat{y}$ should be in the range between 0 and 1. To achieve that, a sigmoid function (σ) should be applied, see Equation (Equation 5).
(5)$$\hat{y}=\sigma \left(w^{T}x+b\right)$$

#### 2.3.2. Random Forest

Random forest (RF) is based on an ensemble for trees. Each tree depends on a collection of random variables. For, a p-dimensional random vector $X={\left(X_{1},X_{2},\ldots ,X_{p}\right)}^{T}$ represents the predictor variables and a real-valued response *Y* (random variable), the joint distribution $P_{XY}\left(X,Y\right)$ is assumed to be unknown. A prediction function f(x) is needed to be obtained to predict *Y*. The loss function L(Y,f(x)) can be used to determine the prediction function. The prediction function is defined to minimize the expected value of the loss, as shown in Equation (Equation 6).
(6)$$E_{XY}\left(L\left(Y,f\left(x\right)\right)\right)$$

#### 2.3.3. Support Vector Machines

A special function called the kernel is the most important aspect of support machines (SVM). The kernel converts the experimental data set from its space into a higher dimensional space where the algorithm constructs a hyperplane that separates between classes. On both sides of the separating plane, there are two parallel hyperplanes that must be constructed. The borders of classes are defined by these hyperplanes. The more distance between these parallel hyperplanes, the higher the accuracy of the SVM algorithm [27]. In this study, a linear kernel was used in the SVM model.

#### 2.3.4. Multi Layer Perceptron

Multi-layer perceptrons (MLPs) can approximate any continuous function, instead of approximating only linear functions [28]. MLPs are composed of several neurons that are organized in at least 3 layers:An input layer that receives the input features and distributes them to the first hidden layer.One or more hidden layers. The input of the first hidden layer is the features distributed by the input layer, while the input of the other hidden layers is the output of each perceptron from the previous layer.One output layer of perceptrons.

In this work, a MLP with one input layer (size 12 perceptrons), two hidden layers (size 24, 6 perceptrons), and one output layer (size 2 perceptrons) was used. The connections between the neurons have weights that are adjusted during the learning process using the backpropagation algorithm.

#### 2.3.5. XGBoost

XGBoost stands for extreme gradient boosting which is an open-source library that provides a framework for regularized gradient boosting. The goal of the XGBoost’s project is to provide a library that is scalable, portable, and featured with distributed gradient boosted decision tree. XGBoost implements gradient boosting, which an ensemble learning approach combines the results from several decision trees for the creation of prediction scores [29]. Characteristics of XGBoost include: a proportional shrinking of leaf nodes, clever penalization of trees, newton boosting, implementation on single and distributed systems, and extra randomization parameter.

## 3. Experimental Results

The performance of our system in the first stage using Algorithm 1 is presented in Table 2 and using Algorithm 2 is presented in Table 3. The performance of our system in the second stage using Algorithm 1 is presented in Table 4 and using Algorithm 2 is presented in Table 5. For 10 different iterations, the dataset was randomly divided into a training set, validation set, and testing set with proportions that are equal to 50%, 25%, and 25%, respectively. In each iteration, we calculate the classification accuracy; then, we estimate the standard deviation for the ten values of accuracy. Features importance is estimated using XGBoost classifier (the classifier of the best accuracy across the five tested machine learning models) and SHAP library [30], as shown in Figure 2. The first six features presented in the figure are the same most important features across the five machine learning models.

Figure 3 and Figure 4 show how the classification accuracy of the classifiers changes according to the number of the selected features in the first and second stages, respectively. A receiver operating characteristic (ROC) curve is a graphical plot that illustrates the diagnostic ability of a binary classifier system as its discrimination threshold is varied. Figure 5 shows the ROC curves for the optimal machine learning models for classification stage 1 and classification stage 2.

Table 2 and Table 3 show that the first-stage performance of the five tested machine learning methods using Algorithm 1 and Algorithm 2, respectively. It can be noticed that all classifiers have acceptable performance. When the filter-based method is the feature selection scheme, Table 2 shows that the XGBoost classifier results in the best performance. SVM and RF result in the same classification accuracy. However, SVM has an advantage over RF as the former used fewer features (only 6) to achieve the same performance. Notably that MLP used only three features to achieve 80% accuracy. The three features were RR, PaO2/FiO2, and LDH. We can conclude that these features have the most discriminative power in differentiating class 0 from other classes. To find whether there are significant differences in the first-stage classification accuracy, when the filter-based method is the feature selection scheme, paired t-tests between XGBoost (the classifier with the highest accuracy) and LR, RF, SVM, and MLP are performed. The *p*-values are 0.002, 0.009, 0.01, and 0.005, respectively. The *p*-values indicate that the classification accuracy is statistically improved using XGBoost classifier over other classifiers. When the PCA method is the feature reduction scheme, Table 3 shows that the XGBoost classifier also achieves the best performance. SVM and MLP classifiers result in similar accuracy. However, MLP uses the smallest number of dimensions resulting from the PCA. To find whether there are significant differences in the first stage classification accuracy, when the PCA method is the feature reduction scheme, paired t-tests between XGBoost (the classifier with the highest accuracy) and LR, RF, SVM, and MLP are performed. The *p*-values are 0.007, 0.009, 0.04, and 0.02, respectively. The *p*-values indicate that the classification accuracy is statistically improved using XGBoost classifier over other classifiers.

Table 4 and Table 5 show the second-stage performance of the five tested machine learning models using Algorithm 1 and Algorithm 2, respectively. It can be noticed that all classifiers have acceptable performance. In Table 4, XGBoost results in the best performance, similar to the first-stage results. SVM, and MLP result in a similar performance in terms of classification accuracy, however, MLP has an advantage over the SVM as the former uses fewer features (only 5) to achieve the same performance. To find whether there are significant differences in the second-stage classification accuracy, when the filter-based method is the feature selection scheme, paired t-tests between XGBoost (the classifier with the highest accuracy) and LR, RF, SVM, and MLP are performed. The *p*-values are 0.006, 0.017, 0.035, and 0.037, respectively. The *p*-values indicate that the classification accuracy is statistically improved using XGBoost classifier over other classifiers. Similarly, Table 5 show that the XGBoost classifier achieves the best performance over other classifiers, similar to the first-stage results. To find whether there are significant differences in the second-stage classification accuracy, when the PCA method is the feature reduction scheme, paired t-tests between XGBoost (the classifier with the highest accuracy) and LR, RF, SVM, and MLP are performed. The *p*-values are 0.007, 0.005, 0.016, and 0.03, respectively. Similar to the results of the first-stage, the *p*-values indicate that the second-stage classification accuracy is statistically improved using XGBoost classifier over other classifiers.

## 4. Discussion

There is certainly some degree of variability in when clinicians feel the need to begin either supplementary non-invasive ventilatory support and ventilatory support. Having said that, indications for initiation of non-invasive ventilatory support (class 1) are fairly standard across the United States, and include criteria such as a blood oxygen saturation level below 90%, significant increase in the work of breathing (these include clinical signs like stridor, use of accessory muscles of ventilation, increased respiratory rate, and the more subjective factor of “oxygen hunger”, which is a perception on the patient’s part that they are struggling to breath). The last element, which is a subjective feature, does introduce a greater element of variability in who receives class 1 ventilatory support, but other elements are more objective. Invasive ventilation, which involves intubation and mechanical ventilation, is more objective still, as it is indicated when patients fail to maintain blood oxygen saturations at or above 90% (hypoxemic respiratory failure) or who are developing progressive retention of carbon dioxide in the blood stream (hypercapnic respiratory failure) despite maximum non-ventilatory support.

The advantage of machine learning is not so much in helping clinicians decide whom to intubate or place on supplemental oxygen—they already have a good idea of when to do that. Rather, we are using the degree of ventilatory support as a surrogate for the severity of COVID-19 disease. As such, clinicians could use this model to predict who will decline most significantly, and it is these patients who should receive more aggressive treatment with antivirals like Paxlovid or intravenous immunoglobulin, both of which are precious resources.

To aid the clinicians, this study proposes a two-stage system for predicting the required level of respiratory support for COVID-19 patients. In this study, different feature selection/reduction methods are investigated. Feature selection/reduction reduces the number of features before training the models. This helps to reduce overfitting, improve accuracy, and reduce training time. In addition, five machine learning classifiers have been tested. The statistical analysis, performed in this study, indicate that the classification accuracies of both the first and second stages are statistically improved using XGBoost classifier over other classifiers XGBoost is a powerful classifier that works well in the studied classification problem [18,21], thanks to its powerful implemented ensemble learning characteristic.

In the first and the second stages, the best achieving models used three important features that have a strong discriminative power. The features are PaO2/FiO2, RR, and LDH. The first one describes the number of breaths you take per minute. The second one describes the ratio of arterial oxygen partial pressure to fractional inspired oxygen. The third one describes a non-specific biomarker of infection, damaged heart muscle, skeletal muscle, or red blood cells. The three most important features obtained using Algorithms 1 and 2 are the same features resulted from using SHAP library. XGBoost uses six features to achieve 84% classification accuracy in the first stage. These features are PaO2/FiO2, RR, LDH, Lymphocytes, CRP, Neutrophils. Lymphocyte is a type of white blood cells, CRP is the level of C-reactive protein, which increases when there is inflammation inside the body and Neutrophils is the ratio of neutrophils to lymphocytes, which describes the balance between systemic inflammation and immunity. As it is expected, the most important two features among these six feature are related to the respiratory system (PaO2/FiO2 and RR).

Despite the fact that our system is the first study to differentiate between three levels of support, we provide here a rough comparison between the performance of our first stage (minimal support vs others) and other methods presented in the literature. Note that the comparison is not fair, since it is done on different data. However, it can give a rough estimate of the potential of the first stage of the proposed system, compared to other methods. Table 6 shows a comparison between the proposed methods and others. The only system that results in a better area under the curve (0.97) than the proposed system is the work of Bendavid et al. [22] However, the proposed approach uses a much higher number of patients (3491 vs. 1061).

Based on the significant research on COVID-19’s clinical course and pathophysiology, adding variables such as age, biological sex, and medical comorbidities, particularly comborid cardiac, pulmonary, and metabolic conditions, may well enhance the predictive accuracy of the system. While such work is currently underway, we cannot at present comment on the exact added value of such data. Furthermore, when patients present de novo to a hospital emergency department, there is often limited information on their exact past medical history and comorbidities. Thus, we feel there is a value in a system that has good predictive accuracy like PaO2/FiO2, respiratory rate, and LDH. Limitations of this study include the use of a single dataset from one center. Another limitation is that all of the instances are recorded at a single time point.

## 5. Conclusions

During COVID-19 pandemic, hospitals face considerable challenges regarding hospital resources, including the required ventilation. Therefore, a system that provides clinicians with the required level of respiratory support for each patient is of great importance, to assign resources to patients who are in need. In this study, a machine learning-based system is proposed for predicting the required level of respiratory support for COVID-19 patients. The system is constructed in two stages; a classification stage between class 0 (minimal support) and others, then a classification stage between class 1 (non-invasive support) and 2 (invasive support). To support the proposed system, different feature selection methods are investigated, namely, ANOVA and recursive feature eliminations, and compared with a PCA dimensionality reduction approach. The results show that the ANOVA feature selection method yields the best results. In addition, five machine learning classifiers have been tested, where the XGBoost classifier achieves the best performance for both the two classification stages. Compared to other systems, the proposed system achieves competing accuracy and is the first one to automatically differentiate between three levels of respiratory support. Therefore, it may help in delivering the care properly to COVID-19 patients. Future work should include system validation on external datasets from different centers, testing its performance on data that are collected at different time points, and investigating the effect of the addition of medical comorbidities, age, and sex, as input to the system. In addition, the proposed system’s generalizability to solve other classification problems will be investigated.

## Figures and Tables

**Figure 1 bioengineering-09-00536-f001:**
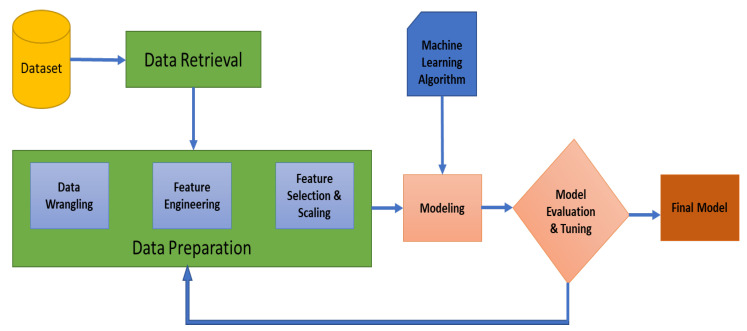
Schematic illustration of our proposed framework.

**Figure 2 bioengineering-09-00536-f002:**
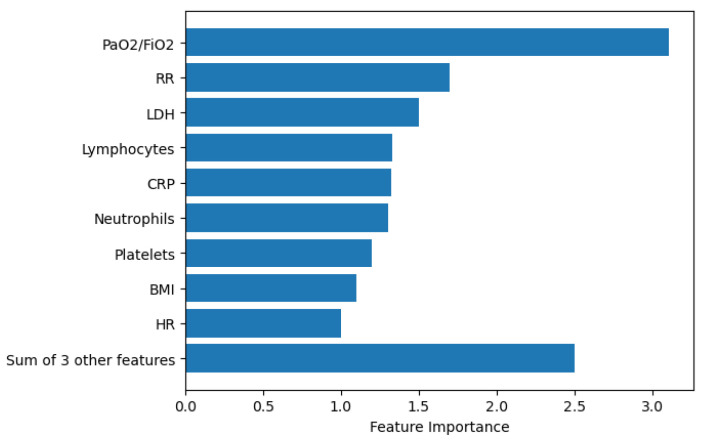
Features importance using XGBoost and SHAP library [30].

**Figure 3 bioengineering-09-00536-f003:**
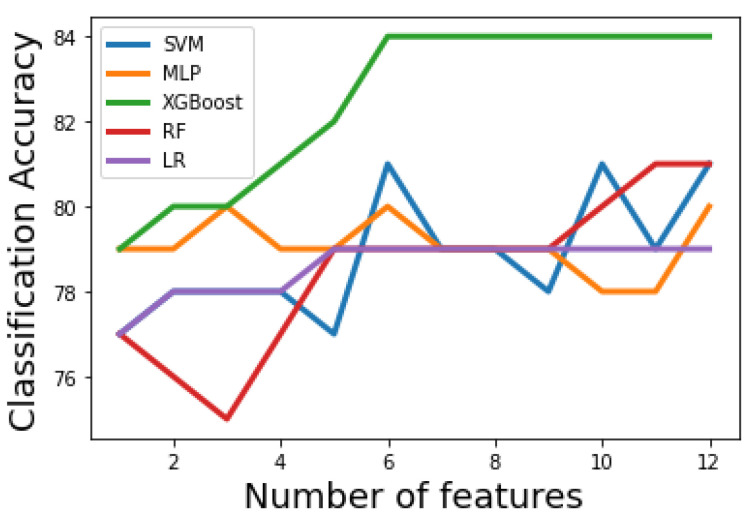
Accuracy curves for the machine learning models in stage 1. XGBoost classifier results in optimal accuracy.

**Figure 4 bioengineering-09-00536-f004:**
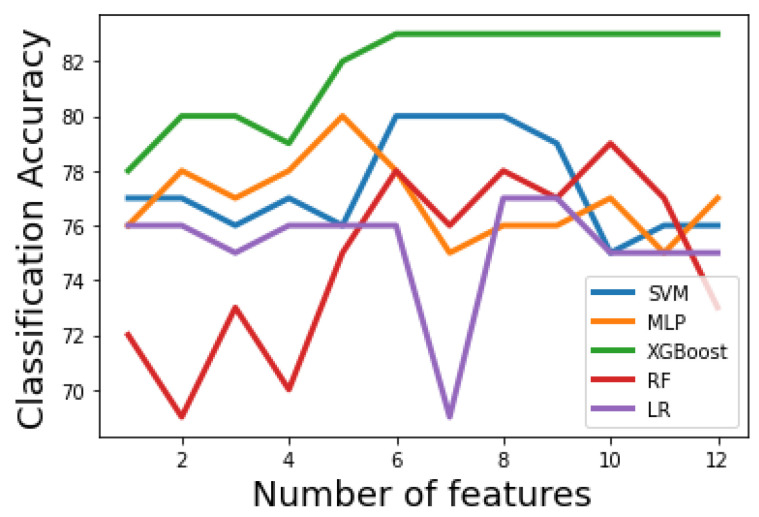
Accuracy curves for the machine learning models in stage 2. XGBoost classifier results in optimal accuracy.

**Figure 5 bioengineering-09-00536-f005:**
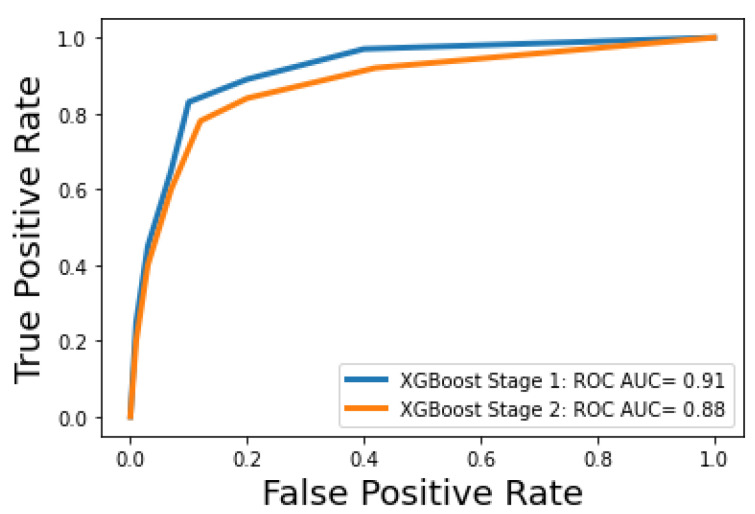
ROC curves for the optimal machine learning model in stage 1 and stage 2.

**Table 1 bioengineering-09-00536-t001:** The mean and standard deviation of the 12 features across the three classes.

	Class 0	Class 1	Class 2
BMI (kg/$m^{2}$)	29.97±7.27	30.38±7.17	32.06±6.98
MAP (mmHg)	86.24±16.33	86.84±16.89	82.57±19.04
HR (beats/min)	94±21.04	95.89±20.16	101.46±25.96
T (C)	37.34±0.82	37.57±0.95	37.59±1.11
RR (breath/min)	21.42±5.59	24.02±6.58	27.13±8.14
PaO2/FiO2 (mmHg)	339.89±106.80	269.53±102.20	238.49±110.11
Platelets (×1000/uL)	223.73±87.09	218.21±86.43	216.11±95.50
Lymphocytes (×1000/uL)	1.31±0.73	1.04±0.58	1.08±0.90
Neutrophil/Lymphocyte	5.90±3.33	6.16±3.41	7.37±3.81
CRP (mg/L)	52.65±56.00	65.17±67.04	99.18±713.6
LDH (units/L)	585.26±233.93	645.54±310.75	697.65±362.10
D-dimer (ng/mL)	2374.75±8382.66	2088.48±5252.43	3680.06±13141.4

**Table 2 bioengineering-09-00536-t002:** Quantitative evaluation of the first stage of our system when filter-based method is the feature selection scheme. The values are average accuracy ± standard deviation.

	LR	RF	SVM	MLP	XGBoost
Accuracy	79%±0.35	81%±0.54	81%±0.33	80%±0.39	84%±0.22
Sensitivity	76%±0.73	80%±0.61	79%±0.41	76%±0.25	82%±0.45
Specificity	81%±0.38	82%±0.42	83%±0.15	82%±0.68	85%±0.19
# features	5	11	6	3	6

**Table 3 bioengineering-09-00536-t003:** Quantitative evaluation of the first stage of our system when PCA method is the feature reduction scheme. The values are average accuracy ± standard deviation.

	LR	RF	SVM	MLP	XGBoost
Accuracy	80%±0.64	80%±0.28	81%±0.31	81%±0.68	83%±0.24
Sensitivity	79%±0.77	78%±0.45	79%±0.58	79%±0.51	81%±0.64
Specificity	81%±0.41	80%±0.56	83%±0.18	83%±0.24	84%±0.19
# dimensions	11	12	10	8	7

**Table 4 bioengineering-09-00536-t004:** Quantitative evaluation of the second stage of our system when filter-based method is the feature selection scheme. The values are average accuracy ± standard deviation.

	LR	RF	SVM	MLP	XGBoost
Accuracy	77%±0.68	79%±0.65	80%±0.64	80%±0.35	83%±0.51
Sensitivity	74%±0.39	79%±0.22	80%±0.37	78%±0.64	81%±0.77
Specificity	78%±0.42	80%±0.41	81%±0.55	80%±0.54	83%±0.22
# features	8	10	6	5	6

**Table 5 bioengineering-09-00536-t005:** Quantitative evaluation of the second stage of our system when PCA method is the feature reduction scheme. The values are average accuracy ± standard deviation.

	LR	RF	SVM	MLP	XGBoost
Accuracy	77%±0.37	77%±0.41	79%±0.62	80%±0.55	81%±0.23
Sensitivity	77%±0.19	75%±0.52	78%±0.31	79%±0.29	80%±0.45
Specificity	79%±0.31	79%±0.33	81%±0.71	80%±0.48	82%±0.18
# dimensions	10	11	9	6	7

**Table 6 bioengineering-09-00536-t006:** Quantitative Comparison between our method and other methods in the literature. AUC: area under the curve.

Method	AUC	Number of Patients
Noupour et al. [15]	0.90	482
Kabbaha at al. [16]	0.72	1613
Zeidberg et al. [18]	0.81	1621 (training) + 1122 (test)
Burdick et al. [20]	0.87	197
Bolourani et al. [21]	0.77	11,525
Bendavid et al. [22]	0.97	1061
Proposed system	0.91	3491

## Data Availability

Data are available upon reasonable request to the corresponding author.

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
