# Peer review of "Predicting the Level of Respiratory Support in COVID-19 Patients Using Machine Learning"

_bioengineering, 2022, doi:10.3390/bioengineering9100536_

Round 1

Reviewer 1 Report

Overall, the paper considers an interesting topic of high importance for medical professionals. However, the article requires significant adjustments to be made prior to being considered for publishing. Authors should refrain from using first person and a narrative manner of writing, and should follow a scientific writing style. Secondly, the article is structured as it is common for a research article, so it is not necessary to state the next section and the content of the article in the text.

As COVID-19 impacts chronic respiratory patients more severely, you should put into perspective the machine learning algorithms used for prediction and classification of respiratory diseases such as:

(2018). An expert diagnostic system to automatically identify asthma and chronic obstructive pulmonary disease in clinical settings. Scientific reports, 8(1), 1-9.

    (2015). Neuro-fuzzy classification of asthma and chronic obstructive pulmonary disease. BMC medical informatics and decision making, 15(3), 1-9.

As there is a lot of research done when it comes to the application of AI for COVID-19, your introduction should have a stronger literature review that will be used in the discussion section for comparison.

The methodology section describes the statistical approach in significant detail, however, the explanation of machine learning is very scarce and lacks the explanation of features of the ML algorithms that are used further in the paper.

The results section should begin with the performance results of your classifiers, everything before belongs to the methodology section. The results section should be organizes in a manner that every table and figure is preceeded with an explanation on what the table/figure presents and followed by a discussion of the presetned results. The discussion should contain a comparison of the results with other relevant results (preferrably the studies that you presented in the introduction section). Additionally you should discuss the limitations to your study both in comparison with other studies and individually.

The conclusion should reiterate the most important results and present the future perspectives of the work and put them into perspective with other relevant findigs and sress out the most important scientific contribution of the resulst presented.

Author Response

We sincerely appreciate the valuable comments, suggestions, and feedback provided by the reviewer on our manuscript. We thank you, in advance, for your careful consideration of our point-by-point responses given below. The revised version has been updated carefully to address all comments raised by the reviewer. Our point-wise response is provided below and is also reflected in the manuscript with yellow highlighted text.

  • Overall, the paper considers an interesting topic of high importance for medical professionals. However, the article requires significant adjustments to be made prior to being considered for publishing. Authors should refrain from using first person and a narrative manner of writing, and should follow a scientific writing style. Secondly, the article is structured as it is common for a research article, so it is not necessary to state the next section and the content of the article in the text.

Thank you for your constructive feedback and comments. The paper was modified to remove the first person and narrative manner of writing.

  • As COVID-19 impacts chronic respiratory patients more severely, you should put into perspective the machine learning algorithms used for prediction and classification of respiratory diseases such as:

(2018). An expert diagnostic system to automatically identify asthma and chronic obstructive pulmonary disease in clinical settings. Scientific reports, 8(1), 1-9.

(2015). Neuro-fuzzy classification of asthma and chronic obstructive pulmonary disease. BMC medical informatics and decision making, 15(3), 1-9.

Thank you for your comment. Several previous studies were added ([13],[14],[15],[16], and [17]), including the mentioned studies ([14] and [13]) in the Introduction section. Please refer to page 2, the second paragraph of the revised manuscript.

  • As there is a lot of research done when it comes to the application of AI for COVID-19, your introduction should have a stronger literature review that will be used in the discussion section for comparison.

Thank you for your comment. Several previous studies were added to the Introduction section, please refer to Page 2, second paragraph. The proposed approach was compared to previous studies in the Discussion section, please refer to the last paragraph on page 11 (Discussion section) and Table 6 on Page 12.

  • The methodology section describes the statistical approach in significant detail, however, the explanation of machine learning is very scarce and lacks the explanation of features of the ML algorithms that are used further in the paper.

Thank you for your comment. A description of the used machine learning models was added to the Methods section. Please refer to Page 6-7, sections 2.3.1, 2.3.2, 2.3.3, 2.3.4, and 2.3.5, which describe in detail the features of the five ML algorithms that are used further in the paper.

  • The results section should begin with the performance results of your classifiers, everything before belongs to the methodology section. The results section should be organizes in a manner that every table and figure is preceeded with an explanation on what the table/figure presents and followed by a discussion of the presetned results. The discussion should contain a comparison of the results with other relevant results (preferrably the studies that you presented in the introduction section). Additionally you should discuss the limitations to your study both in comparison with other studies and individually.

Thank you for your suggestions and comments. In the revised manuscript, the Results and Discussion sections were modified as suggested and became separated from each other. The modified Results Section now begins with the performance results of the proposed classifiers (everything that was before was moved to the proper location in the Materials and Methods Section) and is organized such that every table and figure is preceded with an explanation of what the table/figure presents. A separate section, Discussion, followed the Result section, contains a detailed discussion of the presented results, a comparison of the results with other relevant results, and discusses the limitations of the proposed study.  Please refer to the updated Experimental Results section on pages 8-10, and the updated Discussion section on pages 11-12.

  • The conclusion should reiterate the most important results and present the future perspectives of the work and put them into perspective with other relevant findigs and sress out the most important scientific contribution of the resulst presented.

Thank you for your comment. The Conclusions Section is updated to follow the suggested comments. Please refer to Page 12, the Conclusions Section, in the revised manuscript.

Reviewer 2 Report

The authors proposes an ML-driven system for predicting level of respiratory support. There are several shortcoming: 

1. It is unclear why the author uses a 2-level ML-system instead of using ordinal classification to predict between minimal, non-invasive and invasive systems. A bi or multi-level system typically has less accuracy and less explainability compared to a. single-level system. At least, the authors should compare with a single level ordinal classification. There are quite a few ordinal classification packages in python that allows one to do so. 

2. The authors use PCA etc for feature selection. It is unclear why it is even necessary when using random forests or MLP. Random forests are very good at only using necessary features, and hardly need feature selection. Same to some extent with MLPs. 

3. Given there are so many different types of MLPs, the author should give much more details on MLPs. Also it is unclear why XGBoost is not used when it seems to be one of the state-of-the-art ML algorithms and the authors also cite works where it is used. 

4. The authors should use Shapley to provide more explainability about the features that are useful across algorithms. Are the same features given same importance based on shap values across algorithms, or are they different ? Further, overall Shapley importance values for features can also be used to determine which features are useful. 

Author Response

We sincerely appreciate the valuable comments, suggestions, and feedback provided by the reviewer on our manuscript. We thank you, in advance, for your careful consideration of our point-by-point responses given below. The revised version has been updated carefully to address all comments raised by the reviewer. Our point-wise response is provided below and is also reflected in the manuscript with yellow highlighted text.

  • It is unclear why the author uses a 2-level ML-system instead of using ordinal classification to predict between minimal, non-invasive and invasive systems. A bi or multi-level system typically has less accuracy and less explainability compared to a. single-level system. At least, the authors should compare with a single level ordinal classification. There are quite a few ordinal classification packages in python that allows one to do so. 

Thank you for this helpful comment. In the revised manuscript, we have compared the proposed two-level system with a single-level ordinal classification. Initially, one-stage classification, using similar machine learning algorithms, was investigated, and built using Python classification packages. However, a low classification accuracy of 73% was obtained. Therefore, a two stages system was used for classification which improved the accuracy. This explanation was added to the revised manuscript. Please refer to the first paragraph on Page 3, lines 3-6. 

  • The authors use PCA etc for feature selection. It is unclear why it is even necessary when using random forests or MLP. Random forests are very good at only using necessary features, and hardly need feature selection. Same to some extent with MLPs. 

Thank you for your comment. Feature selection reduces the number of features before training the models. This helps to reduce overfitting, improve accuracy, and reduce training time. Please, see the third paragraph (lines 2-5) on page 11 (Discussion Section).

  • Given there are so many different types of MLPs, the author should give much more details on MLPs. Also it is unclear why XGBoost is not used when it seems to be one of the state-of-the-art ML algorithms and the authors also cite works where it is used. 

Thank you so much for this valuable comment that leads to improve the readability as well as the results of the paper. A subsection (2.3.4) that describes the details of the MLP was added to the Material and Methods Section, Page 7. Following your suggestion, we have tested using an XGBoost classifier against other tested machine learning models. Hopefully, the XGBoost achieves statistically better performance than all other tested machine learning models, thanks to its powerful implemented ensemble learning characteristic and your very helpful suggestion. Please refer to subsection 2.3.5 on page 7, Tables 2 to 5, page 8, the statistical analysis on page 10, and the third paragraph (lines 6-11) on page 11 (Discussion section).

  • The authors should use Shapley to provide more explainability about the features that are useful across algorithms. Are the same features given same importance based on shap values across algorithms, or are they different ? Further, overall Shapley importance values for features can also be used to determine which features are useful. 

Thank you for this valuable comment. We have used SHAP library and have found that the first six features presented in Figure 2 (page 9, in the revised manuscript) have the same importance across the five machine learning models. Please refer to the first paragraph of the Experimental Results Section, lines 7-10, page 8, and Figure 2 on Page 9.

Reviewer 3 Report

The authors’ manuscript takes on a significant and urgent problem – developing a predictive model that can improve the treatment of respiratory illnesses by potentially making earlier and more accurate determinations of what patients will require respiratory support. The authors specifically look at Covid-19, but this approach could have general applicability (although it would require different training data of course).

In general, the paper is clearly explained and understandable. However, in my view, it is too limited to stand as a journal paper (as opposed to conference paper or poster), and additional work is required to revise the paper, which I outline below.

As an initial matter, when I did a quick Pubmed search, however, I did find a few recent articles that may be relevant for being cited – these are likely too recent to have been found by the authors in preparing their draft since it is a fast-moving field. I would suggest reviewing the most recent literature to see if anything else should be cited. The papers I found in my search are linked here:

https://pubmed.ncbi.nlm.nih.gov/35999913/

https://pubmed.ncbi.nlm.nih.gov/35929952/

https://pubmed.ncbi.nlm.nih.gov/36051480/

One key point that the authors should address: Have the data been published elsewhere? If not, then there needs to be a more enhanced description of patient recruitment and how clinical data were collected in supplemental materials. I also did not see an ethics statement. Where there inclusion / exclusion criteria for patients? How many hospitals? How many records/hospital? What were patient demographics? Would using other patient features such as BMI, comorbidities, or age/sex change predictions? Is there a reason for including or excluding such variables? These issues need to be addressed in the Methods, and perhaps included in revised analyses.

A table showing the quantitative values for features and their means/standard errors in patients in the 3 classes or something like that might also help for context.

Another point that I did not see for necessary context: What is the usually accepted accuracy for conventional determinations of whether ventilation is needed, i.e., based on clinical protocols or other criteria used by doctors? It’s important to show how ML methods compare to that to demonstrate the extent to which they are helpful. For example, in the US there are ventilator allocation guidelines that are detailed and specific, and I imagine there is similar in other parts of the world. How would ML predictions compare to existing standards? What is the benefit of using ML?

There should be an improved Discussion of the features that were found to be significant by the classifiers and why they are as expected or whether they were unexpected. Do other features not contribute anything? If so can they be excluded in a model? The authors should compare a model with a reduced number of features to one with more features.

Also, the work relies entirely on cross-validation, but there may be cohort effects that result in a higher accuracy by cross-validation than would be justified.  Also, the Method section does not explain, unless I missed it, what the outcome here was – was it the ultimate allocation of ventilator usage? If it was, then you might expect, for example, that when cases are very high, for ventilators to be allocated in fewer instances than when cases are low. Or, the characteristics of patients might influence ventilator usage, or the originating hospital. One way of checking against cohort effects would be to use hold-out group analysis: for example, using time-separate training/test sets, e.g., training data up to April 2021, and testing on data after April 2021 (or whatever separation makes sense). These and other issues should also be discussed in the Discussion section.

Finally, while the authors look at RF, they do not consider LightGBM and/or XGBoost, which should be considered as well. I would also point out that these methods would not require imputation for missing values. A revised paper should compare one or both of these methods, which are easy to implement. Also, as a minor point, the Methods section should explain how the method was implemented (e.g., in Python? sklearn?) with appropriate citations.

Author Response

We sincerely appreciate the valuable comments, suggestions, and feedback provided by the reviewer on our manuscript. We thank you, in advance, for your careful consideration of our point-by-point responses given below. The revised version has been updated carefully to address all comments raised by the reviewer. Our point-wise response is provided below and is also reflected in the manuscript with yellow highlighted text.

  • The authors’ manuscript takes on a significant and urgent problem – developing a predictive model that can improve the treatment of respiratory illnesses by potentially making earlier and more accurate determinations of what patients will require respiratory support. The authors specifically look at Covid-19, but this approach could have general applicability (although it would require different training data of course).

Thank you very much for this positive attitude towards our work. Although the current system is tested on COVID data classification, we plan to check its generalizability to solve other classification problems. Please, see the last three lines in the Conclusions Section, page 12.

  • In general, the paper is clearly explained and understandable. However, in my view, it is too limited to stand as a journal paper (as opposed to conference paper or poster), and additional work is required to revise the paper, which I outline below.

Thank you for your comment. Major extensive modifications to all parts of the manuscript (the Introduction, Material and Methods, Experimental Results, Discussion, and Conclusions Sections) have been performed to the manuscript to stand as a journal paper. These modifications include:

  1. Improving the literature review by adding more related methods (second paragraph, page 2)
  2. Including more information about the data description (section 2.1, page 3) and data features (Table 1, page 5)
  3. Detailed illustration of the Machine learning models (section 2.3, pages 6 and 7)
  4. Introducing the XGBoost classifier to the tested machine learning classifiers  (section 2.3.5, page 7) and updating the Results and Discussion Sections to include XGBoost classifier results and their discussions (Table 2 to 5), page 8.
  5. Including an analysis of feature importance using SHAP library (page 8, first paragraph of the Experimental Results Section) and Figure 2 on page 9.
  6. Including ROC analysis for the best classifier on both two-stages of the proposed system (second paragraph on age 8, and Figure 5 on page 10)
  7. Including a separate section, Discussion, followed the Result section, contains a discussion of the presented results, a comparison of the results with other relevant results, and discusses the limitations of the proposed study.  Please refer to the Discussion Section on pages 11-12.
  8. In the revised manuscript, we have performed statistical analysis to find whether there are significant differences in the performance of the ML models. We added four types of statistical analysis and added them to the revised manuscript (The two paragraphs on page 10, Experimental Results Section).
  9. The conclusion is updated to reiterate the most important results and present the future perspectives of the work and put them into perspective with other relevant findings and stress the most important scientific contribution of the results presented (Conclusion section, page 12).

  • As an initial matter, when I did a quick Pubmed search, however, I did find a few recent articles that may be relevant for being cited – these are likely too recent to have been found by the authors in preparing their draft since it is a fast-moving field. I would suggest reviewing the most recent literature to see if anything else should be cited. The papers I found in my search are linked here:

https://pubmed.ncbi.nlm.nih.gov/35999913/

https://pubmed.ncbi.nlm.nih.gov/35929952/

https://pubmed.ncbi.nlm.nih.gov/36051480/

Thank you for your comment. Several previous studies were added ([13],[14],[15],[16], and [17]), including the mentioned studies ([15], [16], and [17]) in the Introduction section. Please refer to page 2, second paragraph, on the revised manuscript.

  • One key point that the authors should address: Have the data been published elsewhere? If not, then there needs to be a more enhanced description of patient recruitment and how clinical data were collected in supplemental materials. I also did not see an ethics statement. Where there inclusion / exclusion criteria for patients? How many hospitals? How many records/hospital? What were patient demographics? Would using other patient features such as BMI, comorbidities, or age/sex change predictions? Is there a reason for including or excluding such variables? These issues need to be addressed in the Methods, and perhaps included in revised analyses.

Thank you for raising these issues. They certainly merit clarification. These data have not been published elsewhere. We have added the following text to the manuscript:

"Retrospective data were collected by the Center of Excellence in Respiratory Infectious Diseases at the University of Louisville starting in February 2020, and the data analyzed here extend from the creation of the dataset through December 31, 2020. Eight different hospitals in the Louisville metropolitan area were included, constituting one academic medical center, two other tertiary care hospitals, and five community hospitals in the area. Inclusion criteria were a positive COVID-19 polymerase chain reaction test, symptomatic infection, and admission to the hospital. Symptoms were defined as any respiratory symptoms, diarrhea, or those of systemic sepsis. The only exclusion criterion was age less than 18. The UofL Institutional Review Board (IRB) has approved the acquisition and data collecting; IRB number 21.0673. The study complied with the Helsinki Declaration.” 

--Please, refer to the first paragraph of Section 2.1, page 3 of the revised manuscript

Body mass index (BMI) was indeed used as an input in this system. Medical comorbidities, age, and sex were not. We have forthcoming work that will also use these variables as inputs. We certainly think that based on prior knowledge of COVID-19 pathophysiology that these are likely worthwhile variables to include, but we cannot as yet comment precisely on their exact added value. We have added a comment on this into the discussion:

"Based on the significant research on COVID-19's clinical course and pathophysiology, adding variables such as age, biological sex, and medical comorbidities, particularly comborid cardiac, pulmonary, and metabolic conditions, may well enhance the predictive accuracy of the system. While such work is currently underway, we cannot at present comment on the exact added value of such data. Furthermore, when patients present de novo to a hospital emergency department, there is often limited information on their exact past medical history and comorbidities. Thus, we feel there is a value in a system that has good predictive accuracy like PaO2/FiO2, respiratory rate, and LDH."

--Please refer to the first paragraph on page 12, Discussion section 

A table showing the quantitative values for features and their means/standard errors in patients in the 3 classes or something like that might also help for context.

Thank you for your suggestion. We added Table 1, which contains the means and standard deviations of the used features across the three classes. Please refer to the first three lines on page 5 and Table 1, Page 5 (subsection 2.2.2). 

  • Another point that I did not see for necessary context: What is the usually accepted accuracy for conventional determinations of whether ventilation is needed, i.e., based on clinical protocols or other criteria used by doctors? It’s important to show how ML methods compare to that to demonstrate the extent to which they are helpful. For example, in the US there are ventilator allocation guidelines that are detailed and specific, and I imagine there is similar in other parts of the world. How would ML predictions compare to existing standards? What is the benefit of using ML?

There is certainly some degree of variability in when clinicians feel the need to begin either supplementary non-invasive ventilatory support and ventilatory support. Having said that, indications for initiation of non-invasive ventilatory support (class 1) are fairly standard across the United States, and include criteria such as a blood oxygen saturation level below 90%, significant increase in the work of breathing (these include clinical signs like stridor, use of accessory muscles of ventilation, increased respiratory rate, and the more subjective factor of "oxygen hunger", which is a perception on the patient's part that they are struggling to breath). The last element, which is a subjective feature, does introduce a greater element of variability in who receives class 1 ventilatory support, but other elements are more objective. Invasive ventilation, which involves intubation and mechanical ventilation, is more objective still, as it is indicated when patients fail to maintain blood oxygen saturations at or above 90% (hypoxemic respiratory failure) or who are developing progressive retention of carbon dioxide in the blood stream (hypercapnic respiratory failure) despite maximum non-ventilatory support.

The advantage of ML is not so much in helping clinicians decide whom to intubate or place on supplemental oxygen - they already have a good idea of when to do that. Rather, we are using the degree of ventilatory support as a surrogate for the severity of COVID-19 disease. As such, clinicians could use this model to predict who will decline most significantly, and it is these patients who should receive more aggressive treatment with antivirals like Paxlovid or intravenous immunoglobulin, both of which are precious resources.

–Please, refer to the first and second paragraphs of the Discussion Section, on page 11.

  • There should be an improved Discussion of the features that were found to be significant by the classifiers and why they are as expected or whether they were unexpected. Do other features not contribute anything? If so can they be excluded in a model? The authors should compare a model with a reduced number of features to one with more features.

Thank you for your comment. We estimated features' importance across all classifiers. Please refer to Figure 2 (page 9) and the first paragraph (lines 7-10) of the Experimental Results section, page 8. 

As it is expected, the most important two features (PaO2/FiO2 and respiratory rate) are related to the respiratory system. Nothing was unexpected. We exclude unnecessary features from the model by performing feature selection. Our experiments investigate the use of different numbers of features and we report the best performing model and the used number of features, in the best case which is XGBoost, uses a reduced number of features. We also plotted model performance during the use of different numbers of features in Figures 3 and 4. An analysis of the feature importance is added to the Discussion Section (Please, refer to the fourth paragraph in the Discussion section, page 11). 

  • Also, the work relies entirely on cross-validation, but there may be cohort effects that result in a higher accuracy by cross-validation than would be justified.  Also, the Method section does not explain, unless I missed it, what the outcome here was – was it the ultimate allocation of ventilator usage? If it was, then you might expect, for example, that when cases are very high, for ventilators to be allocated in fewer instances than when cases are low. Or, the characteristics of patients might influence ventilator usage, or the originating hospital. One way of checking against cohort effects would be to use hold-out group analysis: for example, using time-separate training/test sets, e.g., training data up to April 2021, and testing on data after April 2021 (or whatever separation makes sense). These and other issues should also be discussed in the Discussion section.

Thank you for your comment. We do not do cross-validation. The data splitting goes as follows: For 10 different iterations, the dataset was randomly divided into a training set, validation set, and testing set with proportions that are equal to 50%, 25%, and 25%, respectively. In each iteration, we calculate the classification accuracy, then we estimate the standard deviation for the ten values of accuracy. Please refer to the updated manuscript on page 6, the first paragraph of Section 2.3, lines 3-5. 

The ultimate goal of the study is to help clinicians in decision-making regarding the ventilation requirements of COVID-19 patients. We added this clarification to the Discussion Section. Furthermore, regarding the use of time-separate training/test sets, we discussed this in the Discussion and Conclusion:  Limitations of our study include the use of a single dataset from one center. Another limitation is that all of our instances are recorded at a single time point. Future work might include the validation of the developed approach on an external dataset from different centers and the validation of data collected at different times from the training set. Please, check the first paragraph in the Discussion Section, page 11, the Last paragraph in the Discussion Section, page 12 (last three lines) and the future work, and the end of the Conclusions section (last five lines).

  • Finally, while the authors look at RF, they do not consider LightGBM and/or XGBoost, which should be considered as well. I would also point out that these methods would not require imputation for missing values. A revised paper should compare one or both of these methods, which are easy to implement. Also, as a minor point, the Methods section should explain how the method was implemented (e.g., in Python? sklearn?) with appropriate citations.

Thank you so much for this valuable comment that leads to improve the results of the paper. Following your suggestion, we have tested using an XGBoost classifier against other tested machine learning models. Hopefully, the XGBoost achieves statistically better performance than all other tested machine learning models, thanks to its powerful implemented ensemble learning characteristic and your suggestion. Please refer to subsection 2.3.5 on page 7, Tables 2 to 5, page 8, the statistical analysis on page 10, and the first three lines of the last paragraph of the Discussion section, page 11. We implemented all the methods using Python and scikit-learn. Please refer to the first paragraph in the Materials and Methods section (last three lines), page 3.

Round 2

Reviewer 2 Report

The authors have sufficiently improved the manuscript in response to the first round of comments. 

Reviewer 3 Report

Thank you for implementing my comments and suggestions; the manuscript is much stronger now.